

# Antimicrobial and antibiofilm activity of specialized metabolites isolated from *Centaurea hyalolepis*

Shurooq Ismail[1,2], Marco Masi[1], Rosa Gaglione[1,3], Angela Arciello[1,3] and Alessio Cimmino[1]

[1] University of Naples Federico II, Naples, Italy
[2] An-Najah National University, Nablus, Palestine
[3] Istituto Nazionale di Biostrutture e Biosistemi, Rome, Italy

## ABSTRACT

The discovery of plant-derived compounds that are able to combat antibiotic-resistant pathogens is an urgent demand. Over years, *Centaurea hyalolepis* attracted considerable attention because of its beneficial medical properties. Phytochemical analyses revealed that *Centaurea* plant species contain several metabolites, such as sesquiterpene lactones (STLs), essential oils, flavonoids, alkaloids, and lignans. The organic extract of *C. hyalolepis* plant, collected in Palestine, showed significant antimicrobial properties towards a panel of Gram-negative and Gram-positive bacterial strains when the Minimal Inhibitory Concentration (MIC) values were evaluated by broth microdilution assays. A bio-guided fractionation of the active extract *via* multiple steps of column and thin layer chromatography allowed us to obtain three main compounds. The isolated metabolites were identified as the STLs cnicin, 11β,13-dihydrosalonitenolide and salonitenolide by spectroscopic and spectrometric analyses. Cnicin conferred the strongest antimicrobial activity among the identified compounds. Moreover, the evaluation of its antibiofilm activity by biomass assays through crystal violet staining revealed almost 30% inhibition of biofilm formation in the case of *A. baumannii* ATCC 17878 strain. Furthermore, the quantification of carbohydrates and proteins present in the extracellular polymeric substance (EPS) revealed the ability of cnicin to significantly perturb biofilm structure. Based on these promising results, further investigations might open interesting perspectives to its applicability in biomedical field to counteract multidrug resistant infections.

# INTRODUCTION

Finding healing powers in plants is an ancient idea. Plants have always represented an important source of therapeutic compounds and have inspired the production of synthetic or biotechnological drugs, including antimicrobial compounds. Recently, several efforts have been also devoted to the discovery of plant-derived compounds, which are able to counteract the worrying phenomenon of the spread of antibiotic-resistance (*Machado et al., 2023*). Antimicrobial resistance (AMR) is the ability of microorganisms to adapt and

Corresponding authors
Angela Arciello, anarciel@unina.it
Alessio Cimmino, alessio.cimmino@unina.it

thrive in the presence of compounds previously able to influence their survival (*Bottery, Pitchford & Friman, 2021*). AMR is a worrying phenomenon threatening human health that, if not resolved, might significantly reduce the effectiveness of conventional antibiotics against bacterial infections (*Álvarez-Martínez, Barrajón-Catalán & Micol, 2020*). Bacteria are able to adapt to the presence of antibiotics by mutations, horizontal gene transfer or modifications of gene expression. Once antibiotic resistance is acquired, it could be transmitted to future bacterial generations either by cell division or by horizontal gene transfer (*Christaki, Marcou & Tofarides, 2020*). Antibiotic abuse in health sector, agricultural practices, together with the high rate of mobility of people, animals, and goods around the world have greatly contributed to the spread of antimicrobial resistance in recent years (*Bottery, Pitchford & Friman, 2021*). Considering the dearth of discovery of new antimicrobial drugs, the phenomenon has emerged as a grave threat to current and future medical practice and socioeconomic activity, as recently highlighted by numerous organizations and governments (*Sulis, Sayood & Gandra, 2022*). In this scenario, the development of effective alternatives to conventional antibiotics has become urgent and medicinal plants might represent a potential source of novel drugs.

A microbial community of cells adhered to an abiotic or a biotic surface, which is made of complex extracellular polymeric substance (EPS) composed primarily of proteins, polysaccharides, and eDNA is known as biofilm (*Gandhi et al., 2017*). The formation of bacterial biofilms is particularly due to interactions of filamentous bacteria, microbial aggregates, inorganic and organic particles, which are held together by EPS (*Palomares-Navarro et al., 2023*). Furthermore, the biofilm serves as a barrier to invasions and offers resistance towards protozoans, grazers, and host immunological responses (*Gandhi et al., 2017*). The formation of bacterial biofilm is also responsible for the ineffectiveness of conventional antibiotics (*Cangui-Panchi et al., 2023*). Indeed, the three-dimensional network of the EPS matrix protects bacteria from the effects of disinfectants, thus making biofilms difficult to eradicate (*Flemming et al., 2016*). Therefore, it is necessary to investigate and to explore new anti-biofilm compounds. In the last years, there has also been a growing application of biochemical and molecular biology methods to the isolation and purification of compounds with specific and potentiated activities. This requires several steps, such as extraction, fractionation, purification, characterization, and evaluation of biological activity and of putative cytotoxic effects (*Cook et al., 2023*).

*Centaurea hyalolepis* is among the plants that attracted attention for their therapeutic properties. It belongs to the well-known Asteraceae, also named Compositae family (*Danin, 2018*), one of the biggest plant families comprising several genera and more than twenty-three thousand species of shrubs, herbs, and trees (*Danin, 2018*; *Djeddi, Argyropoulou & Skaltsa, 2008*). The species belonging to the *Centaurea* genus are predominately distributed in West Asia and Mediterranean area (*Tiwana et al., 2021*). Since the phytochemical constituents of plants represent a precious reservoir of novel, diverse and potent drugs (*Chemsa et al., 2018*), researchers devoted several efforts to the investigation of wild plants endowed with beneficial medical properties. *Centaurea* species have been widely used as a remedy for several health-related issues, such as diarrhea, urinary tract infections, diuretic problems, diabetes, inflammation, bacterial infections, dandruff, and others (*Chemsa*
*et al., 2018*; *Zengin et al., 2010*; *Csupor et al., 2010*; *Shoeb et al., 2005*). Phytochemical analyses revealed that *Centaurea* plant species contain several metabolites, such as sesquiterpene lactones (STLs), essential oils, flavonoids, alkaloids, and lignans (*Khammar & Djeddi, 2012*; *Ismail et al., 1986*). It is worth noting that STLs and flavonoids are the predominant metabolites of *Centaurea* (*Maos, Anastácio & Nunes, 2021*), and they are probably responsible for most of the reported biological activities of *Centaurea* crude extracts (*Khammar & Djeddi, 2012*).

Recently, STLs were widely analyzed for their biological properties, and found to confer antibacterial, antifungal, anti-inflammatory, anti-cancer, and anti-oxidative properties (*Laurella et al., 2022*). This elicited a great interest that led to the purification and isolation of STLs from Asteraceae family members which are known to be very rich in STLs (*Picman, 1986*). It was demonstrated that the unique bioactivity of STLs is correlated to their molecular structure characterized by the presence of $\alpha, \beta$-unsaturated carbonyl groups with $\alpha$-methylene or $\gamma$-lactone rings aside by the basic 15-carbon backbone structure (*Picman, 1986*; *Kupchan et al., 1970*). The molecular structure of STLs might allow covalent interactions with sulfhydryl groups of enzymes or other nucleophilic biological molecules (*Kupchan et al., 1970*; *Schmidt, 1999*). In this study, we found that *C. hyalolepis* crude extracts showed significant antimicrobial properties towards Gram-negative and Gram-positive bacterial strains, such as *Staphylococcus aureus* ATCC 29213, methicillin-resistant *Staphylococcus aureus* MRSA WKZ-2, *Enterococcus faecalis* ATCC 29212, *Escherichia coli* ATCC 25922, *Salmonella enterica* subsp. *enterica* Serovar Typhimurium ATCC 14028, and *Acinetobacter baumannii* ATCC 17878. Since dichloromethane ($CH_2Cl_2$) organic extract exhibited the strongest antibacterial activity with respect to other extracts obtained in different solvents, it was used as starting point to perform a bio-guided purification of *C. hyalolepis* extract. At the end of each step of the purification process, the obtained fractions were tested for their antimicrobial properties to select those containing plant metabolites responsible for the observed antibacterial activity. This manuscript reports for the first time the antimicrobial activity of *C. hyalolepis* plant extract and the isolation of three main metabolites by bio-guided purification as well as a significant antibiofilm activity of cnicin towards the tested bacterial strains. Compounds were identified by spectroscopic and spectrometric analyses and their stereochemistry was confirmed by comparing the optical rotation values with literature data.

## MATERIALS & METHODS

### Materials

A Jasco P-1010 digital polarimeter (Tokyo, Japan) was used to measure optical rotations. [1]H NMR spectra were recorded at 500 MHz in CDCl$_3$ and CD$_3$OD on a Bruker spectrometer and the same solvents were used as internal standards. Electrospray ionization mass spectra (ESIMS) were obtained as previously described (*Ismail et al., 2023*). Column chromatography (CC) was performed on silica gel (Kieselgel 60, 0.063–0.200 mm; Merck, St. Louis, MO, USA). Preparative and analytical thin-layer chromatography (TLC) procedures were performed as previously described (*Zorrilla et al., 2023*). CC silica gels

were from Merck (St. Louis, MO, USA), Kieselgel 60, 0.063–0.200 mm. All the reagents and the solvents were purchased from Merck (St. Louis, MO, USA), unless specified otherwise.

## Bacterial strains and growth conditions

Six bacterial strains were used to evaluate the antibacterial properties of the plant extracts, fractions, and pure compounds. Tested bacterial strains include *Staphylococcus aureus* ATCC 29213, methicillin-resistant *Staphylococcus aureus* MRSA WKZ-2, *Enterococcus faecalis* ATCC 29212, *Escherichia coli* ATCC 25922, *Salmonella enterica* subsp. *enterica* Serovar Typhimurium ATCC 14028, and *Acinetobacter baumannii* ATCC 17878. The same strains were used for the biofilm tests. Bacterial strains were grown in Muller Hinton Broth (MHB; Becton Dickinson Difco, Franklin Lakes, NJ, USA) and on Tryptic Soy Agar (TSA; Oxoid Ltd., Hampshire, UK). In all the experiments, bacteria were inoculated and grown overnight in MHB at 37 °C (*Gaglione et al., 2019*).

## Plant collection and identification

*C. hyalolepis* leaves were collected in West-Bank, Palestine, during March 2021 and identified by Dr. Ghadeer Omar (Department of Biology and Biotechnology at An-Najah National University in Palestine) by using "Field guide to wild flowers of Jordan and neighboring countries Vegetation of Israel and Neighboring Countries" (*Al-Eisawi, 1998*; *Danin, 2018*). A representative plant specimen has been deposited at An-Najah National University herbarium (voucher number ANUH1625). The collected leaves of *C. hyalolepis* were then washed with water, dried, and powdered using a blender prior to the extraction process.

## Plant extraction

To prepare plant extracts, leaves powder (728.0 g) was macerated (1 × 2.5 L) for 48 h in $H_2O$/MeOH (1/1, v/v) under stirring at room temperature. After centrifugation at 7.000 rpm for 40 min, the supernatant was extracted in *n*-hexane (3 × 1.0 L) and successively in $CH_2Cl_2$ (3 × 1.0 L). Methanol was then removed under reduced pressure and the residual water phase extracted by using EtOAc (3 × 500 mL).

## Bio-guided fractionation and identification of isolated compounds

The organic extract obtained in $CH_2Cl_2$ (3.5 g), which showed antimicrobial activity against Gram-negative and Gram-positive bacterial strains (Table 1), was purified by CC performed using as eluent mixture a solution of $CHCl_3$/*i*-PrOH (9/1, v/v) obtaining homogeneous fractions (Fig. 1). The preliminary antimicrobial activity was tested against two bacterial strains, *i.e.,* the Gram-negative *Salmonella enterica* subsp. *enterica* Serovar Typhimurium ATCC 14028 and the Gram-positive *E. faecalis* ATCC 29212. In the case of antimicrobial activity, further analyses were performed against four additional bacterial species, such as *S. aureus* ATCC 29213, methicillin-resistant *S. aureus* MRSA WKZ-2, *E.coli* ATCC 25922, and *A. baumannii* ATCC 17878. Fractions CH.3-CH.5 were found to exert a significant antibacterial activity and were further purified as shown in Fig. 1. In particular, fractions CH.4 and CH.5 were combined into one fraction named CH.6 as they shared the same profile on TLC and showed similar antibacterial properties (Table 2).
**Table 1** MIC$_{100}$ values (mg/mL) determined for *C. hyalolepis* extracts obtained in *n*-hexane, dichloromethane (CH$_2$Cl$_2$), and ethylacetate (EtOAc).[a]

| Bacterial strains | MIC$_{100}$ (mg/mL) | | | |
| --- | --- | --- | --- | --- |
| | *n*-Hexane extract | CH$_2$Cl$_2$ extract | EtOAc extract | Residual water phase |
| *S. aureus* ATCC 29213 | 1 | 0.25 | 1 | >2 |
| *S. aureus* MRSA WKZ-2 | 1 | 0.5 | 1 | >2 |
| *E. faecalis* ATCC 29212 | 2 | 2 | 1 | >2 |
| *E. coli* ATCC 25922 | 2 | 1 | 1 | >2 |
| *S.* Typhimurium ATCC 14028 | 1 | 1 | 1 | >2 |
| *A. baumannii* ATCC 17878 | 2 | 0.25 | 0.5 | >2 |

**Notes.**
[a]Reported data refer to three biological replicates.

The remaining part of fraction CH.3 (341.7 mg) was further purified by CC and eluted in CH$_2$Cl$_2$/MeOH (95/5, v/v). Subsequent reverse-CC was eluted with acetonitrile/H$_2$O (4/6, v/v) and allowed us to obtain a main metabolite (CH.3.2-3) identified as salonitenolide (compound **3**, 81.1 mg). The remaining part of fraction CH.6 (568.1 mg) was firstly purified by CC and eluted with CHCl$_3$/MeOH (9/1, v/v), thus allowing to obtain six homogeneous fractions (CH.6.1–CH.6.6). The remaining part of sub-fraction CH.6.3 (187.6 mg) was further purified by CC and eluted in CH$_2$Cl$_2$/*i*-PrOH (9/1, v/v), thus providing a main metabolite (CH.6.3-4) identified as cnicin (compound **1**, 53.8 mg). The remaining part of the sub-fraction CH.6.2 (163.7 mg) was further purified by CC and eluted with CH$_2$Cl$_2$/*i*-PrOH (9/1, v/v) to obtain a pure sesquiterpene lactone identified as 11β,13-dihydrosalonitenolide (compound **2**, 44.4 mg).

## Antibacterial activity assays

Total extracts and compounds **1–3** were tested to evaluate their antimicrobial activity as previously described (*Pizzo et al., 2018*). Briefly, bacterial cells were diluted to $2 \times 10^6$ CFU/mL in Nutrient Broth (NB; Difco, Becton Dickinson, Franklin Lakes, NJ, USA) along with increasing amounts of each tested extract or compound. In each case, starting from a stock solution, two-fold serial dilutions were prepared according to broth microdilution method. Following drying, the isolated compounds and fractions were quantified by determining their dry weight by using a digital analytical balance. Stock solutions were prepared in DMSO and further diluted to achieve final DMSO concentration lower than 5%. After overnight incubation of bacterial cells with tested compounds, MIC$_{100}$ values were determined as the lowest concentration responsible for no visible bacterial growth. At least three biological replicates were performed for each experiment.

## Bactericidal activity assays

A colony-counting assay (*Wiegand, Hilpert & Hancock, 2008*) was used to determine the minimal bactericidal concentration (MBC) values of isolated compounds. To do this, bacterial aliquots were taken from the wells with no sign of bacterial growth. Samples were then serially diluted in 0.5X NB, plated on TSA (Tryptic Soy Agar) plates, and incubated for 24 h at 37 °C. Following incubation, bacterial colonies were counted, and the MBC

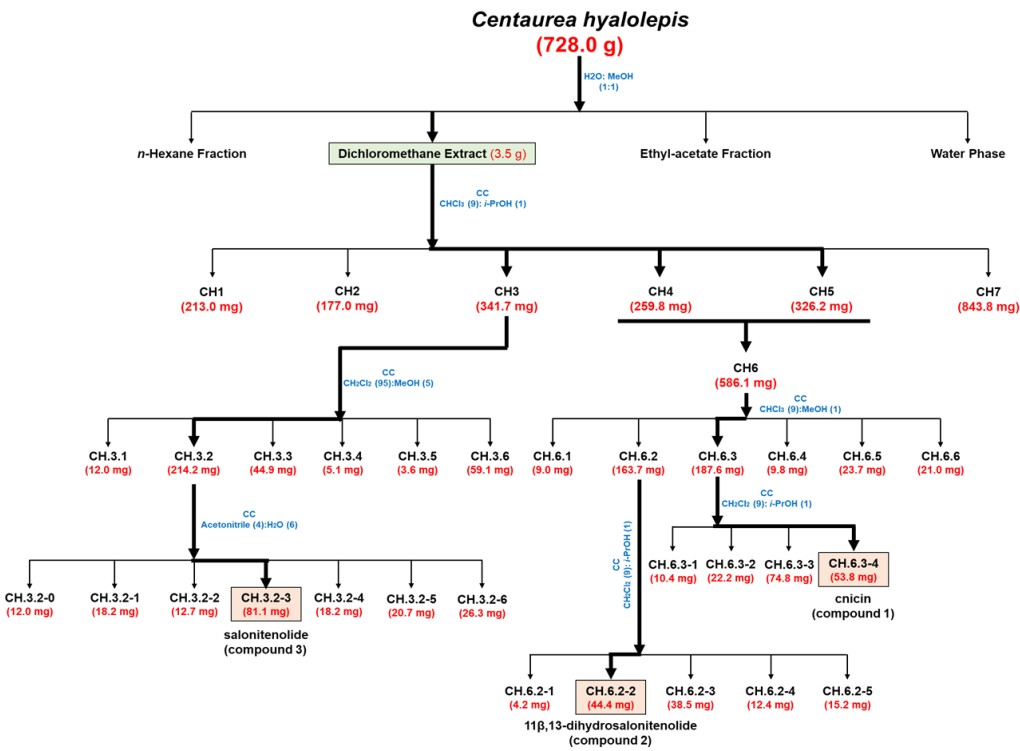

**Figure 1** **Bio-guided fractionation of the extracts.** Schematic overview of the successive bio-guided fractionation steps of the active extract obtained in $CH_2Cl_2$ solvent starting from *C. hyalolepis* leaves ending up with the pure isolated compounds.

**Table 2** **$MIC_{100}$ values (mg/mL) determined for fractions obtained upon purification of *C. hyalolepis* extract obtained in $CH_2Cl_2$ [a].**

| | $MIC_{100}$ (mg/mL) | |
|---|---|---|
| $CH_2Cl_2$ Fraction | *E. faecalis* ATCC 29212 | *S.* Typhimurium ATCC 14028 |
| CH.1 | 0.125 | 0.125 |
| CH.2 | 1 | 1 |
| CH.3 | 0.125 | 0.0312 |
| CH.4 | 0.125 | 0.0625 |
| CH.5 | 0.25 | 0.0312 |
| CH.7 | 1 | 1 |

**Notes.**
[a] Reported data refer to three biological replicates.

values were determined as the lowest compound concentration that resulted in >99.9% cell death relative to the original bacterial inoculum (*Dell'Olmo et al., 2021*). The experiment was performed in triplicate.

## Antibiofilm activity assays

Biomass assays through crystal violet staining were used to evaluate the antibiofilm activity of cnicin (**1**). In particular, bacteria inocula were grown over-night at 37 °C, then diluted

to $2 \times 10^8$ CFU/mL in 0.5X MHB containing increasing concentrations of cnicin (**1**) (0–1 mg/mL) and successively incubated at 37 °C for 24 h. Then the bacterial biofilm was washed three times with phosphate buffer (PBS 1X) and incubated for 20 min with 0.04% crystal violet at room temperature. Samples were washed with PBS and then the dye bound to the cells was dissolved in 33% acetic acid. A wavelength of 630 nm and a microtiter plate reader (FLUOstar Omega, BMG LABTECH, and Germany) were used for spectrophotometric measurements (*Gaglione et al., 2017*). The absorbance values at 630 nm were compared with those obtained for the untreated biofilm sample, thus obtaining the reported percentage of biofilm mass. Three biological replicates were performed for each experiment that was carried out with triplicate determinations.

### Determination of total protein content in biofilm matrix

The effects of compound **1** on the total protein content of biofilm matrix were evaluated on the bacterial strains that were incubated with increasing concentrations of selected compounds (0–1 mg/mL, 1:1 *v/v*). Following incubation, protein content was evaluated by Bradford assay (*Bradford, 1976*). To do this, bacteria were grown overnight in MHB at 37 °C, and then adjusted to $4 \times 10^8$ CFU/mL in a final volume of 100 μL of 0.5X MHB (Difco, Becton Dickinson, Franklin Lakes, NJ) along with increasing concentrations of cnicin (compound **1**) (1:1 *v/v*). The samples were then incubated at 37 °C for 24 h, in order to test compound **1** effects on protein content of formed biofilms. Biofilms were then harvested and suspended in 36 μL of Milli-Q water through ultrasonic oscillation. Afterwards, equal volumes of supernatant solution and Bradford Dye Reagent were mixed at room temperature for 5 min prior to measuring absorbance values at 595 nm (*Guo et al., 2019*). Protein amount was determined by using a standard bovine serum albumin (BSA, 0–2 mg/mL) calibration curve (Fig. S1). Three biological replicates were performed for each experiment.

### Determination of total polysaccharides content in biofilm matrix

The antibiofilm effects of compound **1** were deepened by quantifying the polysaccharides content of biofilm matrix produced in the absence or in the presence of compound **1** by the six bacterial strains used in this study. To do this, phenol-sulphuric acid method was used (*Nielsen, 2010*). Bacteria were grown overnight in MHB at 37 °C, and then adjusted to $4 \times 10^8$ CFU/mL in a final volume of 100 μL of 0.5X MHB (Difco, Becton Dickinson, Franklin Lakes, NJ) along with increasing concentrations of cnicin (compound **1**) (0–1 mg/mL, 1:1 *v/v*). The samples were then incubated for 24 h at 37 °C, to evaluate the effects of compound **1** on polysaccharides content of formed biofilms. Biofilms were then harvested, suspended in 36 μL of Milli-Q water and sonicated for 5min. Afterwards, equal volumes of 5% phenol and five volumes of concentrated $H_2SO_4$ were added to the samples that were incubated at room temperature for 30 min prior to measuring absorbance values at 482 nm (*Guo et al., 2019*). Polysaccharides amount was determined by using a standard glucose (0–5 mg/mL) calibration curve (Fig. S2). Three biological replicates were performed for each experiment.

### Statistical analyses

Statistical analyses were performed by using a student's $t$-test. Significant differences were indicated as $*P < 0.05$, $**P < 0.01$ or $***P < 0.001$. Graphs were performed with the GraphPad Prism 8 software.

## RESULTS

### *C. hyalolepis* leaves extraction, bio-guided fractionation of $CH_2Cl_2$ extract, and antimicrobial activity evaluation

*C. hyalolepis* leaves were extracted as described in Materials and Methods section. In particular, the hydro-alcoholic solution obtained by maceration has been extracted using solvents with increasing polarity, such as *n*-hexane, dichloromethane ($CH_2Cl_2$), and ethyl acetate (EtOAc). Gram-negative (*E. coli* ATCC 25922, *A. baumannii* ATCC 17878, and *Salmonella enterica* subsp. *enterica* Serovar Typhimurium ATCC 14028) and Gram-positive (*S. aureus* ATCC 29213, *S. aureus* MRSA WKZ-2, *E. faecalis* ATCC 29212) bacterial strains were employed to test the activity of the obtained organic extracts by using the broth microdilution assays (*Wiegand, Hilpert & Hancock, 2008*) to determine the minimal inhibitory concentration (MIC) values reported in Table 1.

The $CH_2Cl_2$ extract was found to exhibit the strongest antimicrobial activity against all the tested bacterial strains, with $MIC_{100}$ values ranging from 0.25 to 2 mg/mL (Table 1). Thus, it was selected as a starting point to perform a bio-guided fractionation with the main aim to identify plant metabolites responsible for the observed antibacterial activity. To this purpose, $CH_2Cl_2$ extract was fractionated by CC as reported in Fig. 1 yielding six homogenous fraction groups, which were tested on two bacterial strains, *i.e., Salmonella enterica* subsp. *enterica* Serovar Typhimurium ATCC 14028 and *E. faecalis* ATCC 29212.

These two bacterial strains were selected since they were the most resistant to $CH_2Cl_2$ extract antibacterial activity (Table 1, $MIC_{100}$ values ranging from 2 to 1 mg/mL), thus representing the prototype of a Gram-positive and a Gram-negative bacterial strain suitable to identify the active compounds present in this organic extract. As reported in Table 2, the fractions obtained upon purification by column chromatography were found to be more active than the total $CH_2Cl_2$ extract, being effective on both *E. faecalis* ATCC 29212 and *Salmonella enterica* subsp. *enterica* Serovar Typhimurium ATCC 14028 bacterial strains with $MIC_{100}$ values comprised between 0.0312 and 1 mg/mL (Table 2). Since the two fractions CH.4 and CH.5 were found to share similar chromatographic profiles on TLC, they were combined to form fraction CH.6 which was further fractionated to obtain active metabolites.

### Isolation and identification of pure metabolites from $CH_2Cl_2$ extract

The most active fractions (CH.3-CH.5) were further purified by successive steps of CC (Fig. 1) obtaining three pure metabolites identified by spectroscopic and spectrometric methods ([1]H NMR and ESI MS) as cnicin, 11 β,13-dihydrosalonitenolide, and salonitenolide (compounds **1–3** in Fig. 2). The yields in percentage of compounds **1**, **2**, and **3** were found to be 1.5%, 1.2%, and 2.3%, respectively.

**Figure 2 Isolated compounds.** Chemical structures of cnicin (compound **1**), 11β,13-dihydrosalonitenolide (compound **2**), and salonitenolide (compound **3**).

In particular, the $^1$H NMR and ESI MS data of compound **1** (Figs. S3 and S4) agreed with those reported in the literature for cnicin (*Suchý et al., 1960*). Its absolute configuration was confirmed comparing the specific optical rotation value $[\alpha]^{25}_D = +152.4$ (*c* 0.5, MeOH) with that reported in the literature for this compound (*Kurita et al., 2016*). The structure and absolute stereochemistry of 11β,13-dihydrosalonitenolide (compound **2**) was confirmed by comparing its $^1$H NMR and ESI MS data (Figs. S5 and S6) and the specific optical rotation value $[\alpha]^{25}_D = +81$ (*c* 0.5, CHCl$_3$) with those reported in the literature for the same compound (*Marco et al., 1992*). Finally, compound **3** was identified (Figs. S7 and S8) by comparing its $^1$H NMR and ESI MS data the specific optical rotation value $[\alpha]^{25}_D = +152.4$ (*c* 0.3, MeOH) with those reported in the literature for salonitenolide (*Suchý, Herout & Šorm, 1965*; *Adekenov et al., 1990*).

## Antimicrobial activity of the isolated metabolites

The antibacterial activity of compounds **1–3** was evaluated at different concentrations on *E. coli* ATCC 25922, *S. aureus* ATCC 29213, *S. aureus* MRSA WKZ-2, *E. faecalis* ATCC 29212, *A. baumannii* ATCC 17878, and *Salmonella enterica* subsp. *enterica* Serovar Typhimurium ATCC 14028 bacterial strains. As shown in Fig. 3, cnicin (compound **1**) was found to be the most active compound on all the tested strains. Cnicin was found to be particularly active against Gram-negative bacterial strains *E. coli* ATCC 25922 and *Salmonella enterica* subsp. *enterica* Serovar Typhimurium ATCC 14028 with MIC$_{100}$ values of 0.125 mg/mL corresponding to minimal bactericidal concentration (MBC) values (Fig. 3). Compound **3**, salonitenolide, was found to be effective on all the tested strains even if higher MIC$_{100}$ and MBC values were detected with respect to cnicin (Fig. 3). Compound **2**, 11β,13-dihydrosalonitenolide, showed the lowest antibacterial activity with respect to compounds **1** and **2** with MIC$_{100}$ values ranging from 0.5 to 7.5 mg/mL (Fig. 3). It has also to be highlighted that, in the case of compound **2**, significant differences between MIC$_{100}$ and MBC values were detected on almost all the strains tested (Fig. 3). In general, MIC$_{100}$ values obtained for cnicin were found to be similar or even lower than those detected for the CH$_2$Cl$_2$ extract (Table 1 and Fig. 3). The bacterial strain *E. faecalis* ATCC 29212 was found to be the most resistant to the antimicrobial activity of the tested compounds (Figs. 3 and 4).
### Cnicin Antibacterial Activity

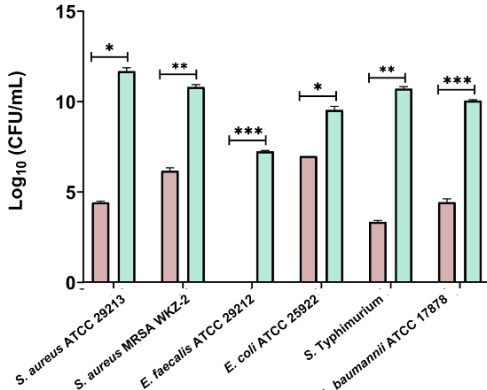

| Bacterial strain | (mg/mL) | |
|---|---|---|
| | $MIC_{100}$ | MBC |
| *S. aureus* ATCC 29213 | 0.25 | 1 |
| *S. aureus* MRSA WKZ-2 | 0.125 | 0.5 |
| *E. faecalis* ATCC 29212 | 1 | 1 |
| *E. coli* ATCC 25922 | 0.125 | 0.25 |
| *S.* Typhimurium | 0.125 | 0.25 |
| *A. baumannii* ATCC 17878 | 0.125 | 1 |

### 11β,13- Dihydrosalonitenolide Antibacterial Activity

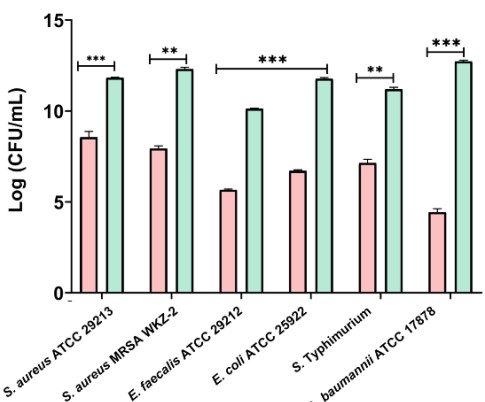

| Bacterial strain | (mg/mL) | |
|---|---|---|
| | $MIC_{100}$ | MBC |
| *S. aureus* ATCC 29213 | 0.5 | 7.5 |
| *S. aureus* MRSA WKZ-2 | 0.625 | 7.5 |
| *E. faecalis* ATCC 29212 | 7.5 | < 7.5 |
| *E. coli* ATCC 25922 | 1.25 | 2.5 |
| *S.* Typhimurium | 0.937 | 7.5 |
| *A. baumannii* ATCC 17878 | 0.5 | 2.5 |

### Salonitenolide Antibacterial Activity

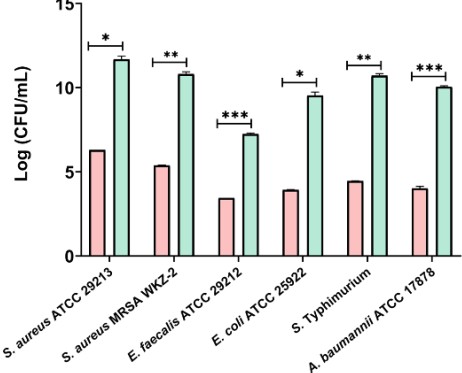

| Bacterial strain | (mg/mL) | |
|---|---|---|
| | $MIC_{100}$ | MBC |
| *S. aureus* ATCC 29213 | 0.25 | 1 |
| *S. aureus* MRSA WKZ-2 | 0.25 | < 1 |
| *E. faecalis* ATCC 29212 | 1* | < 1 |
| *E. coli* ATCC 25922 | 1 | <1 |
| *S.* Typhimurium | 1 | 1 |
| *A. baumannii* ATCC 17878 | 1 | <1 |

*$MIC_{99}$

**Figure 3 Minimal Inhibitory Concentration for compounds 1–3.** Minimal Inhibitory Concentration $MIC_{100}$ (mg/mL) and Minimal Bactericidal Concentration MBC mg/mL) values determined for cnicin (compound **1**) 11β,13-dihydrosalonitenolide (compound **2**) and salonitenolide (compound **3**) tested against a panel of Gram-negative and Gram-positive bacterial strains. Control samples are reported as green bars, whereas treated samples are reported as pink bars. Data represents the mean (±standard deviation, SD) of at least three independent experiments. Significant differences were indicated as $*p < 0.05$, $**p < 0.001$ or $***p < 0.0001$ for treated versus control samples.

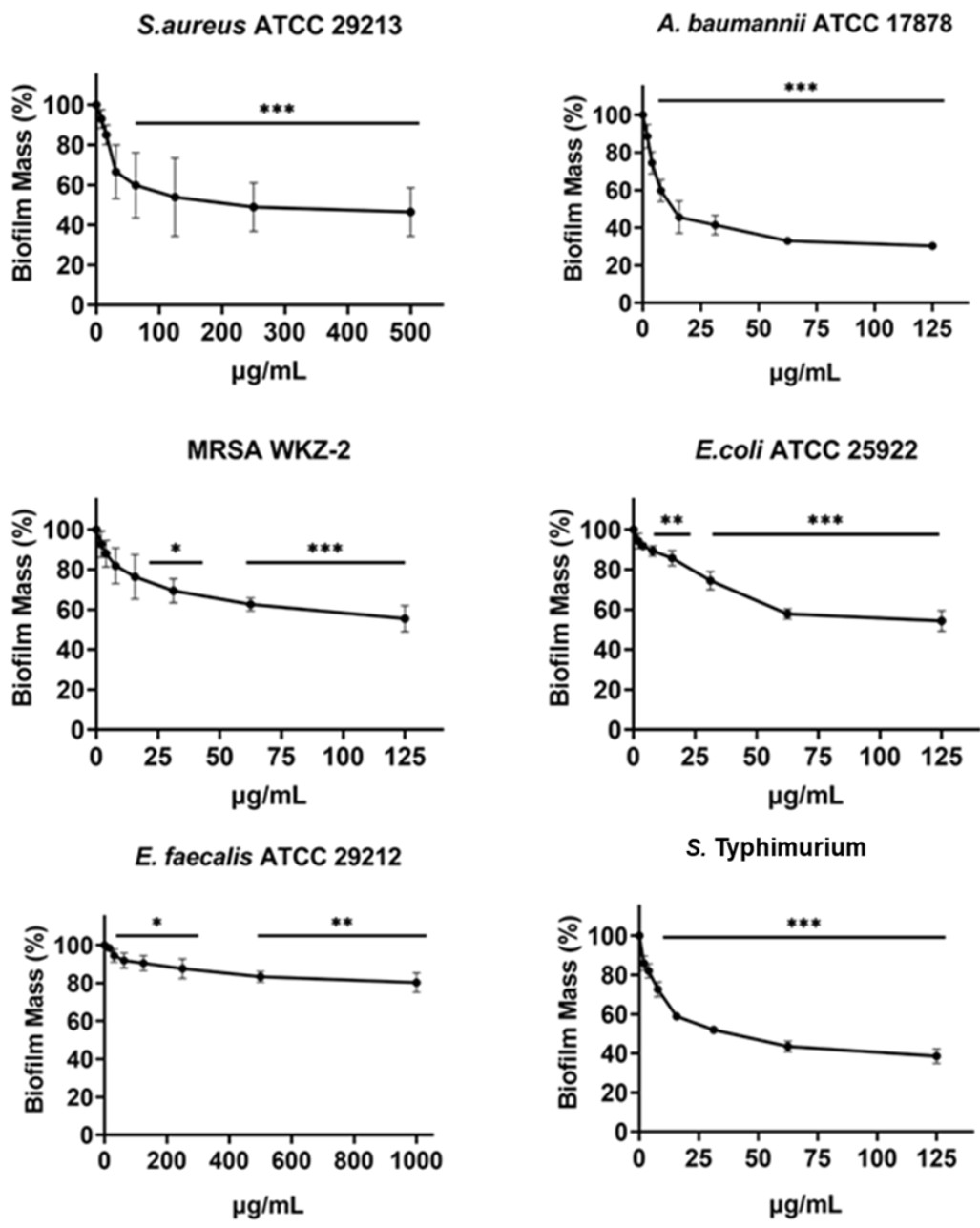

**Figure 4 Antibiofilm activity of increasing concentrations of cnicin (1) evaluated by biomass assays through crystal violet staining on *E. faecalis* ATCC 29212, *A. baumannii* ATCC 17878, *S. aureus* ATCC 29213, *Salmonella enterica* subsp. *enterica* Serovar Typhimurium ATCC 14028, *S. aureus* MRSA WKZ-2, and *E. coli* ATCC 25922.** Crystal violet was used to stain the formed biofilm and samples were analyzed by using a plate reader measuring the absorbance at 630 nm. The evaluation of antibiofilm activity is employed through biomass assays in which the absorbance value at 630 nm of the treated biofilm is divided by the absorbance value at the same wavelength for the untreated biofilm, in this way the percentage of biofilm mass is indicated. Data represent the mean (±standard deviation, SD) of at least three independent experiments. Each experiment was carried out with triplicate determinations. Significant differences were indicated as $*p < 0.05$, $**p < 0.001$ or $***p < 0.0001$ for treated versus control samples.

In the current study it was found that the isolated cnicin and salonitenolide were endowed with significant antibacterial activity against the tested Gram-negative and Gram-positive bacterial strains ($MIC_{100}$ = 0.25–1 mg/mL). 11β,13-Dihydrosalonitenolide (**2**) was found to have a lower antimicrobial activity ($MIC_{100}$ = 0.5–1.25 mg/mL) in comparison with the other two isolated compounds. Among the tested bacterial strains, *E. faecalis* ATCC 29212 was found to be the most resistant to the antimicrobial properties of these metabolites.

### Evaluation of the antibiofilm activity of cnicin (1)

The antibiofilm activity of cnicin (compound **1**) was evaluated by using biomass assays through crystal violet staining on *E. faecalis* ATCC 29212, *S. aureus* ATCC 29213, *S. aureus* MRSA WKZ-2, *A. baumannii* ATCC 17878, *Salmonella enterica* subsp. *enterica* Serovar Typhimurium ATCC 14028 and *E. coli* ATCC 25922 bacterial strains upon incubation with increasing concentrations (Fig. 4). The biofilm formation was significantly inhibited (about 30–50%) for all the tested strains except for *E. faecalis* ATCC 29212. The strongest effects on biofilm formation were observed for *Salmonella enterica* subsp. *enterica* Serovar Typhimurium ATCC 14028 and *A. baumannii* ATCC 17878 (Fig. 4) and significant effects were obtained at concentrations of cnicin (**1**) lower than $MIC_{100}$ values detected on the same bacterial strains (*S. aureus* MRSA WKZ-2, *S. aureus* ATCC 29213, and *E. faecalis* ATCC 29212 in Fig. 3).

### Evaluation of the effects of cnicin (1) on polysaccharides and proteins content in biofilm of tested bacterial strains

Bacterial strains *S. aureus* ATCC 29213, *S. aureus* MRSA WKZ-2, *E. faecalis* ATCC 29212, *A. baumannii* ATCC 17878, *E. coli* ATCC 25922, and *Salmonella enterica* subsp. *enterica* Serovar Typhimurium ATCC 14028 were selected to analyze the effects of cnicin (**1**) on proteins and polysaccharides content in biofilm matrix produced by bacterial cells. To do this, Bradford assay and phenol-sulphuric acid method were used to quantify proteins and polysaccharides, respectively (Figs. 5 and 6). It was demonstrated that the treatment with sub-MIC concentrations of compound **1** induces a significant decrease of polysaccharides' content in all the tested bacterial strains except for *E. faecalis* ATCC 29212 (Fig. 5). Similar results were obtained when the effects of compound **1** were tested on the protein content of biofilm matrix produced by the same bacterial strains. A significant reduction of biofilm protein content was observed upon treatment with sub-MIC concentrations of compound **1** with more pronounced effects in the case of Gram-negative strains with respect to Gram-positive ones (Fig. 6). Further investigations on compound **1** antibiofilm properties revealed significant effects on biofilm mass when the tested bacterial strains were treated with sub-inhibitory doses of cnicin. In particular, the Gram-negative *A. baumannii* ATCC 17878 and *Salmonella enterica* subsp. *enterica* Serovar Typhimurium ATCC 14028 bacterial strains were found to be the most susceptible strains (Fig. 4).

## DISCUSSION

A bio-guided purification of *C. hyalolepis* organic extract obtained in $CH_2Cl_2$ was performed to identify bioactive metabolites responsible for its antimicrobial properties.

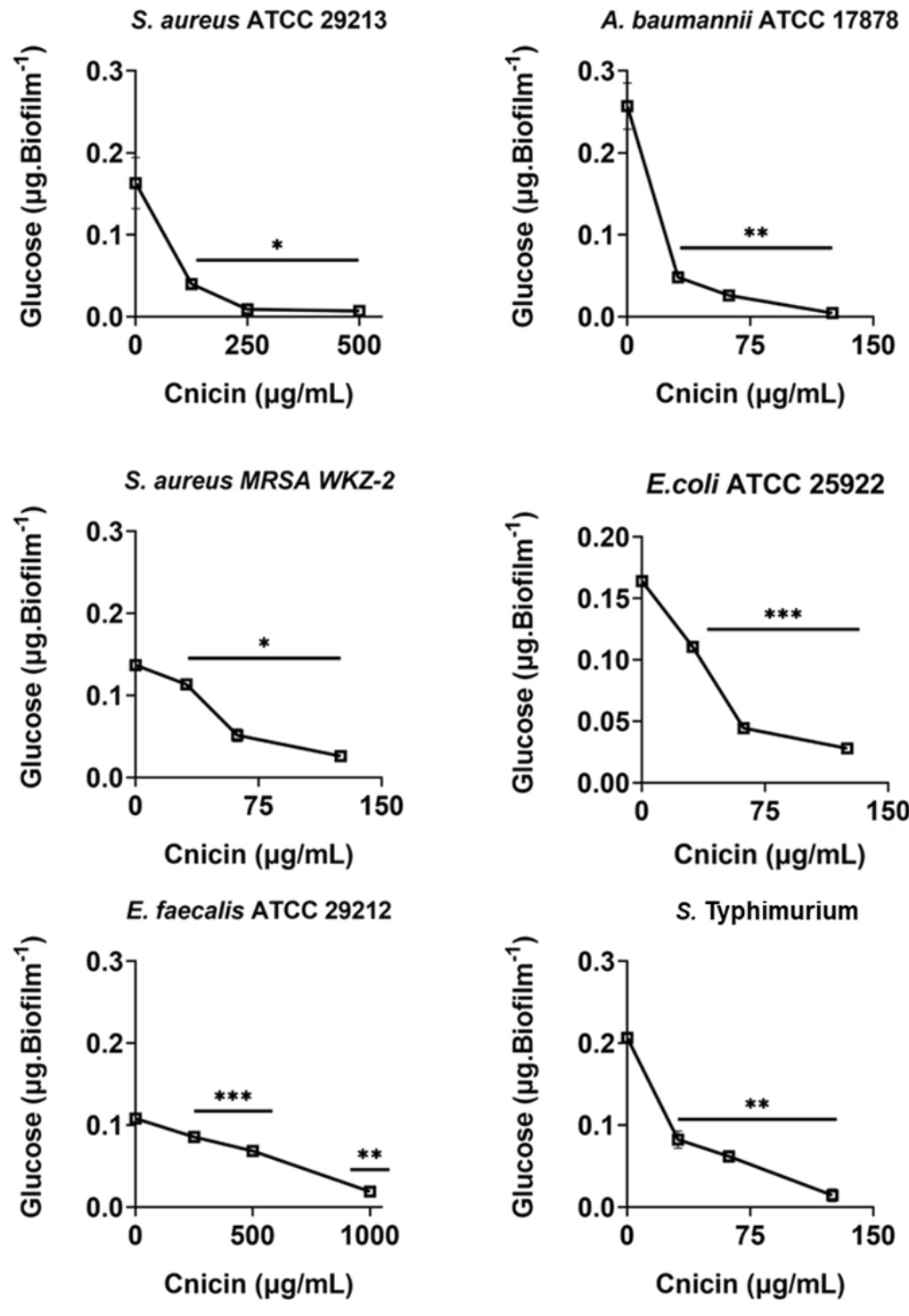

**Figure 5** **Quantification of polysaccharides in biofilm matrix produced through 24 h upon incubation with increasing concentrations of cnicin (1) by phenol-sulphuric acid method on *E. faecalis* ATCC 29212, *A. baumannii* ATCC 17878, *S. aureus* ATCC 29213, *S.* Typhimurium ATCC 14028, *S. aureus* MRSA WKZ-2, and *E. coli* ATCC 25922.** Samples absorbance was measured at 482 nm and data represent the mean (±standard deviation, SD) of at least three independent experiments. Each experiment was carried out with triplicate determinations. Significant differences were indicated as $*p < 0.05$, $**p < 0.001$ or $***p < 0.0001$ for treated versus control samples.

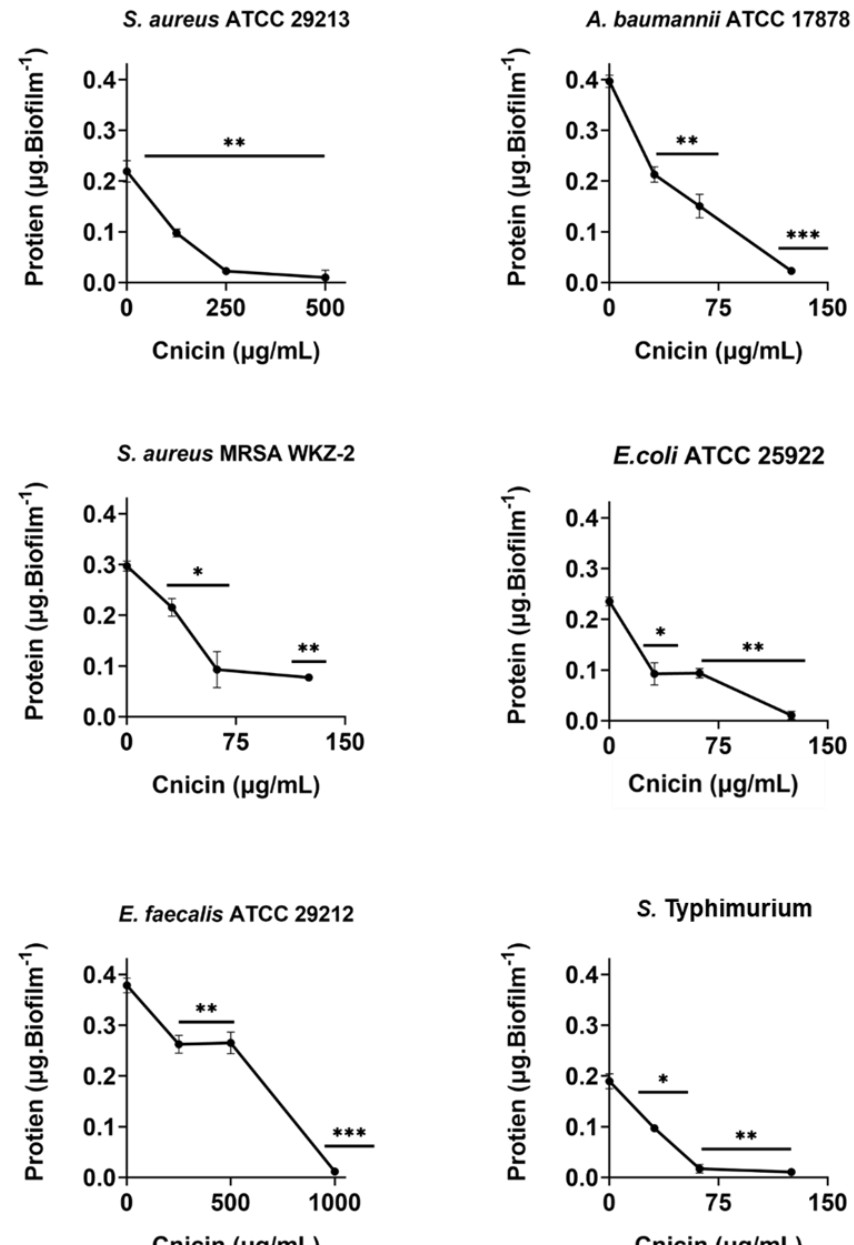

**Figure 6** Quantification of proteins in biofilm matrix produced through 24 h upon incubation with increasing concentrations of cnicin (1) by Bradford assay in on *E. faecalis* ATCC 29212, *A. baumannii* ATCC 17878, *S. aureus* ATCC 29213, *S.* Typhimurium ATCC 14028, *S. aureus* MRSA WKZ-2, and *E. coli* ATCC 25922. Samples absorbance was determined at 592 nm and data represent the mean (±standard deviation, SD) of at least three independent experiments. Each experiment was carried out with triplicate determinations. Significant differences were indicated as *$p < 0.05$, **$p < 0.001$ or ***$p < 0.0001$ for treated versus control samples.

Through the purification process, fractions with antimicrobial activity were analyzed and it was revealed the presence of three sesquiterpene lactones (STLs) identified as cnicin, 11β,13-dihydrosalonitenolide, and salonitenolide (**1–3**). *C. hyalolepis* belongs to the well-known Asteraceae, also named as Compositae family (*Ayad & Akkal, 2019*). Centaurea plant species were reported to be very rich in STLs and flavonoids metabolites (*Ismail et al., 1986*). Cnicin (compound **1**), one of the compounds identified here, was isolated for the first time in 1960 from *Cnicus benedictus* (*Suchý et al., 1960*), a plant that grows in the Mediterranean regions and used for thousands of years to cure liver diseases, anorexia, indigestion problems, ulcers, and swollen fingers (*Berger & Sicker, 2009*). Cnicin was then isolated from several plants (*Suchý et al., 1960*; *Marco et al., 1992*; *Adekenov et al., 1990*; *Ayad & Akkal, 2019*; *Berger & Sicker, 2009*) and analyzed over years to evaluate its beneficial pharmaceutical properties (*Rustaiyan, Niknejad & Aynehchi, 1982*; *Mezache et al., 2010*; *Belkacem et al., 2014*; *Lakhal et al., 2010*; *Kitouni, Benayache & Benayache, 2015*; *Grafakou et al., 2018*; *Fernández, Pedro & Polo, 1995*). The compound **2**, 11β,13-dihydrosalonitenolide, was isolated for the first time from the aerial parts of *C. calcitrapa* together with cnicin (*Marco et al., 1992*). Furthermore, it was isolated from the aerial parts of *C. alba*, *C. spinosa* and *C. pullata* (*Fernández, Pedro & Polo, 1995*; *Saroglou et al., 2005*; *Djeddi, Argyropoulou & Skaltsa, 2008*). Salonitenolide (compound **3**) was isolated for the first time from *C. salonitana* in 1965 (*Suchý et al., 1960*), and then from *Ambrosia artemisiifolia* (*Porter et al., 1970*), *C. malacitana* (*Barrero, Sánchez & Arana, 1988*), and *C. melitensis* (*Ayad et al., 2012*) together with other sesquiterpene lactones, such as cnicin, thus indicating that STLs represent important bioactive compounds of several medicinal plants endowed with anti-inflammatory, anti-diabetic, anti-malarial, anti-proliferative, anti-parasitic, and antibacterial properties (*Salazar-Gómez et al., 2020*).

The isolated cnicin and salonitenolide were found to be endowed with antibacterial activity against the tested Gram-positive and Gram-negative bacterial strains ($MIC_{100}$ = 0.25–1 mg/mL). The compound **2**, 11β,13-dihydrosalonitenolide, was found to have a lower antimicrobial activity ($MIC_{100}$ = 0.5–1.25 mg/mL) in comparison with the other two isolated compounds. Among the tested bacterial strains, *E. faecalis* ATCC 29212 was found to be the most resistant to the antimicrobial properties of these metabolites. Although compounds **1–3** belong to the class of sesquiterpene lactones, key differences in their structural features might be responsible for the detected variations in their antibacterial efficacy (Fig. 2). Moreover, several factors were reported to influence STLs antibacterial properties, such as the compound structure, concentration in solution, hydrophobicity, geometric orientation, the specific bacterial strain tested, and the chemical environment (*Schmidt & Heilmann, 2002*). By comparing the antimicrobial activity of the purified compounds with that of the total extract from *C. hyalolepis*, it appears that the isolated pure compounds were endowed with stronger antibacterial properties. This might be correlated to several factors, such as compounds solubility, pH, iron concentration or antagonistic effects exerted by several metabolites acting in a mixture (*Beigomi, Biabangard & Rohani, 2021*).

Taking into account that cnicin is the strongest antibacterial compound among the three isolated, it is plausible that it mostly contributes to the antibacterial properties of

the whole extract. Cnicin is a STL that belongs to the germacranolide group of terpenes (*Berger & Sicker, 2009*), characterized by an exocyclic α-methylene group in addition to γ-butyrolactone ring (*Berger & Sicker, 2009*). It was demonstrated that this α-methylene group is responsible for the antimicrobial activity of sesquiterpene lactones in general and of cnicin in particular (*Dimkić et al., 2020*; *Adekenov, 1995*). It is worth-noting that cnicin might react through a Michael addition mechanism with nucleophilic groups present in surrounding target molecules, a molecular event that might be at the basis of cnicin antimicrobial activity (*Ismail et al., 1986*). Based on this, the absence of this structural feature in 11β,13-dihydrosalonitenolide (**2**) might be responsible for the significantly lower antimicrobial activity of this compound with respect to that of cnicin under the experimental conditions tested. In the literature, it was reported that cnicin interacts with proteins found in bacterial membranes, thus causing cell lysis and death (*Dimkić et al., 2020*). Cnicin also acts by blocking bacterial cell wall synthesis through the irreversible inhibition of MurA, an enzyme responsible for the catalysis of the first step of the synthesis of peptidoglycans (*Bachelier, Mayer & Klein, 2006*).

Interestingly, cnicin was also found to be endowed with significant antibiofilm properties. Furthermore, a remarkable decrease in the biofilm protein and exopolysaccharide content was evidenced upon treatment with cnicin (Figs. 5 and 6), thus further highlighting the significant antibiofilm properties of compound **1**. The EPS (extracellular polymeric substance) matrix develops a three-dimensional network that shields bacterial cells from antimicrobials and host immune system defense strategies (*Steenackers et al., 2012*). Additionally, the EPS matrix promotes surface adhesion and colonization. Hence, its inhibition or disruption could be considered a key event to prevent or to eradicate biofilms (*Palomares-Navarro et al., 2023*). Previous studies have revealed that the antibiofilm properties of plant-derived compounds, such as phenolics and terpenes, might be correlated to their ability to interfere with EPS production (*Ortega-Ramirez et al., 2020*; *Keelara, Thakur & Patel, 2016*; *Hui et al., 2016*; *Gutierrez-Pacheco et al., 2018*). Moreover, some studies suggested the ability of cnicin to interfere with quorum-sensing (QS), a key step leading to biofilm formation (*Hentzer et al., 2002*). The currently achieved results provide evidence that cnicin is able to prevent biofilm formation, thus opening interesting perspectives to the treatment of antimicrobial resistance (AMR) development.

## CONCLUSION

To the best of our knowledge, this is the first study aimed at identifying metabolites responsible for the antibacterial activity of *C. hyalolepis* collected in Palestine. To do this, the plant dichloromethane (CH$_2$Cl$_2$) organic extract was used as starting point to perform a bio-guided purification of active metabolites. Three sesquiterpene lactones (STLs) were isolated and identified as cnicin, 11β,13-dihydrosalonitenolide, and salonitenolide (compounds **1–3**, respectively). Cnicin was found to be endowed with the strongest antimicrobial activity towards Gram-negative and Gram-positive bacterial strains. Moreover, cnicin also resulted to be a potent antibiofilm agent, thus opening the way to further studies aimed

at deeply characterizing its biological properties and its applicability in biomedical field to counteract multidrug resistant infections.

### Funding

The authors received no funding for this work.

### Competing Interests

The authors declare there are no competing interests.

### Author Contributions

- Shurooq Ismail conceived and designed the experiments, performed the experiments, analyzed the data, prepared figures and/or tables, authored or reviewed drafts of the article, and approved the final draft.
- Marco Masi conceived and designed the experiments, performed the experiments, authored or reviewed drafts of the article, and approved the final draft.
- Rosa Gaglione conceived and designed the experiments, authored or reviewed drafts of the article, and approved the final draft.
- Angela Arciello conceived and designed the experiments, analyzed the data, authored or reviewed drafts of the article, supervised, and administered the research project and contributed materials and reagents, and approved the final draft.
- Alessio Cimmino conceived and designed the experiments, analyzed the data, authored or reviewed drafts of the article, supervised, and administered the research project and contributed materials and reagents, and approved the final draft.

### Data Availability

  The raw data is available in the Supplemental Files.

### Supplemental Information

Supplemental information for this article can be found online at http://dx.doi.org/10.7717/peerj.16973#supplemental-information.

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
