# Peer review of "Antimicrobial and antibiofilm activity of specialized metabolites isolated from Centaurea hyalolepis"

_PeerJ, doi:10.7717/peerj.16973_

## Round 0.1 · original submission · Major Revisions

Please revise the manuscript by following the reviewers' comments. When resubmitting your manuscript, please provide a point-by-point response letter, addressing all the issues sufficiently and satisfactorily.

**Language Note:** The review process has identified that the English language must be improved. PeerJ can provide language editing services - please contact us at [email protected] for pricing (be sure to provide your manuscript number and title). Alternatively, you should make your own arrangements to improve the language quality and provide details in your response letter. – PeerJ Staff

Reviewer 1 ·

Basic reporting

The manuscript describes a very interesting and comprehensive work on the search of new antibacterial compounds. The methods employed are pertinent, and the author performed different chemical procedures to identify the compounds. However, some issues must be cited and modified at the manuscript:

1- First of all, the manuscript should undergo a profound English grammar and structure revision. In several parts of the manuscript the comprehension is difficult as a consequence of several problems with the English structure, and there are several missing words widespread throughout the manuscript.

2- The authors presented their results along with the discussion. Unfortunately, this procedure made the text more dubious and difficult to understand in several parts. I recommend to the author to divide this part of the manuscript, presenting first their results, and after the discussion.

3- Also, the authors included several methodology descriptions in their Results section. This specific situation made the comprehension of this part of the manuscript very difficult. I recommend them to exclude this extensive methodology descriptions from the Results section.

4- The authors listed the references in the same order that thy appeared in the text. Please confirm if the journal´s guide for authors ask to list the references by alphabetical order.

5- There is no need for so many figures at this manuscript. Figures 3, 4, and 5 should be merged. using a similar color scheme for them all.

6- The authors should pay a specific attention to the definition of the figures. I know that PeerJ demands for Figures at PDF, but it was very difficult to understand Figure 1.

7- A special attention should be given to the Discussion, since it is expected that this section must be focused in explain the author´s results using previously published results. In several parts of the discussion, what can be found is only the citation of other researchers results, without linking them to the results obtained by the authors.

Experimental design

8- The authors did not describe in the Methodology section how they did the experiments for the definition of the MBC100. This must be included.

9- The authors used a very inclusive panel of microorganisms. However, there must be a rationale in the inclusion of such species and specific strains. A supplementary figure or table must be also included, showing the resistance profile to conventional antibiotics that have already been described for these strains.

10 - The authors must be aware that they did a procedure to observe the influence of the metabolites in the formation of biofilm by the bacterial strains. Indeed, this is of crucial importance, however it is also needed another experiment, where the authors should determine the activity of the metabolites in the consolidated biofilm. This is important since it can reveal how the metabolites would act in previously established infection, with an already formed biofilm. There are procedures that are already published for such purpose, and they are not time demanding or even costly.

11- In a study like this, the grand finale of the antibiofilm activity definition of an antibacterial compound is represented by a scanning electron microscopy image of the treated biofilm, comparing with images from not treated biofilms. This experiment would be crucial to show the influence of the metabolite treatment at the consolidated biofilm. I would strongly recommend the addition of this experiment.

12 - In broth microdilution assays, it is very important to have a solution of the antimicrobial agent that has its concentration very well defined. It was not clear in the present version of the manuscript how the authors quantified the metabolites at the solutions they used for the broth microdilution assay, and this aspect should be highlighted.

13- Sigma-Aldrich was purchased by Merck, and the headquarters of Sigma are not located in Italy, but in Saint Louis. The authors should correct this kind of issue throughout the manuscript.

14 -Regarding the correct identification of the plant, it is important to cite which botanical guide was used to make this identification, and not the person that made it. It is important to cite what was taken in consideration to correctly identify the plant, which botanical guide or atlas.

15- The range of the dilutions used for the broth microdilution and for the antibiofilm assays should be cited.

16- When analyzing the quantification of proteins and polysaccharides at the biofilm, it was not clear for me how the authors absolutely quantified the biofilm itself, since the results are expressed as ug x biofilm -1. How did they get this quantification? It must be cited.

Validity of the findings

17- The results presented by the authors are valid and interesting. The chemical procedures that they used to identify the metabolites are valid, comprehensive and inclusive. However, the Methodology section must be improved, since it is dubious and lack some important details in many parts.

18- Conclusions are OK. However, I would recommend the exclusion of many phrases that are more focused in the methodology itself, since they can create some confusions while reading the conclusion section.

·

Basic reporting

Reviewer report of Manuscript ID 91636
Congratulations on the present study! However, serious flaws were identified during the revision process and the authors must carefully address these issues for publication endorsement.

General comments
The manuscript is not well-written, and, in my opinion, a major English Editing is needed in the revised manuscript.
The Abstract section must be improved by stating the main results of antibiofilm activity and EPS quantification.
The Introduction section lacks some references and more recent studies. Please see my minor comments below.
The Materials and Methods section is very incomplete, and it must be improved. Please see my minor comments below.
The Result & Discussion section shows tables and figures wrongly explained or without any explanation. Please see my comments below.

Experimental design

Minor comments
Abstract
Line 29-31- “Moreover, evaluating its antibiofilm activity by crystal violet assays as well as by the quantification of carbohydrates and proteins present in the extracellular polymeric substance (EPS) revealed cnicin potent antibiofilm properties.” Please replace crystal violet assays with biomass assays through crystal violet staining and state the biofilm reduction values. In addition, please state the methodology applied to the quantification of carbohydrates and proteins indicating the main results. Finally, it is still unknown to the Readers the range of microorganisms in which the cnicin showed antibiofilm activity.
Introduction
Lines 41-44- “Recently, several efforts have been also devoted to the discovery of plant-derived compounds, which are able to counteract the worrying phenomenon of the spread of antibiotic-resistance (Subramani, Narayanasamy & Feussner, 2017).” The citation is from 6 years ago, so I recommend adding more recent citations. Please consult the following examples:
10.3390/futurepharmacol3030034
10.3390/molecules27227999
Lines 64-66- “Furthermore, the biofilm serves as a barrier to invasions and offers resistance towards protozoan, grazers, and host immunological responses.” Please add the proper citation. Please consult the following example: 10.1016/j.crimmu.2023.100057
Lines 70-73- “In the last years, there has also been a growing application of biochemical and molecular biology methods to the isolation and purification of compounds with specific and potentiated activities, requiring several steps, such as extraction, fractionation, purification, characterization, evaluation of activity and possible cytotoxic effects.” Please properly add citations.
Lines 90-92- “Recently, STLs were widely analyzed for their biological properties, and found to confer antibacterial, antifungal, anti-inflammatory, anti-cancer, and anti-oxidative properties (Chaturvedi & Dwivedi, 2017).” As previously indicated, the citation is from 6 years ago, so I recommend adding more recent citations.
Line 100- “… Gram-negative and Gram-positive bacterial strains” The Readers still do not know which panel of bacteria is evaluated in the present study. Please add it.
Materials and Methods
Lines 139-144- “Their antibacterial activity was tested against two bacterial strains (Gram-negative S. typhimurium ATCC 14028 and Gram-positive E. faecalis ATCC 29212). Indeed, to identify fractions containing antimicrobial compounds, each sample obtained through the purification process was tested towards six bacterial strains, i.e., S. aureus ATCC 29213, E. coli ATCC 25922, methicillin-resistant S. aureus MRSA WKZ-2, E. faecalis ATCC 29212, S. typhimurium ATCC 14028, and A. baumannii ATCC 17878.” The first citation of bacterial names should be written with the full names and then followed by its abbreviation. Please rectify the sentences. Also, “S. typhimurium ATCC 14028” is actually Salmonella enterica subsp. enterica serovar Typhimurium (check: https://www.atcc.org/products/14028) and its abbreviation is S. Typhimurium, in which the serotype or serovar name should not be in italics. Please rectify it and it is important to avoid these mistakes to the non-familiar Readers. Finally, why not just state: “The preliminary antimicrobial activity was realized through two bacterial strains (Gram-negative S. Typhimurium ATCC 14028 and Gram-positive Enterococcus faecalis ATCC 29212) and, if antimicrobial activity was observed, further evaluation was performed in four additional bacterial species, more exactly, Staphylococcus aureus ATCC 29213, methicillin-resistant S. aureus MRSA WKZ-2, Escherichia coli ATCC 25922, and Acinetobacter baumannii ATCC 17878.”
Why split the two S. aureus in the sentence?

Lines 145-147- “In particular, fractions CH.4 and CH.5 were combined into one fraction named CH.6 as they shared the same profile on TLC and showed similar antibacterial properties (Table 2).” In Table 2, there is a fraction called CH.7 but no CH.6. Please rectify it.
Line 157-165- Concerning my previous comment on bacterial full names. This subsection entitled “Bacterial strains and growth conditions” could be moved after the first subsection entitled “Materials”. However, it is only a suggestion.
Lines 172-173- “MIC100 values were determined as the lowest concentration responsible for no visible bacterial growth.” No visible bacterial growth is usually the MIC50 value, it is necessary to apply the formula and OD measurement to state the MIC100 value. In addition, the authors showed MBC values in the Results section and no description of the MBC procedure is given here (it was obtained by resazurin-based 96-well plate microdilution method or growth culture?). Finally, the recommended bacterial inoculum for the microdilution method is 1x10^5 CFU/mL according to CLSI and EUCAST guidelines. Why use 2x106 CFU/mL? There is some explanation that it is not given to the Readers?
Line 175- “Crystal violet assays” Please rectify it. The procedure is called “Biomass assays by crystal violet (CV) staining method”.

Line 212- “Statistical analyses were performed by using a student.s t-test.” Usually, the data set for this type of experiment shows a non-parametric distribution and a non-parametric test should be chosen. Did the authors evaluate the data distribution using the Shapiro–Wilk test?

The Materials and Methods are summarized in two pages and several specifications are missing, such as controls and the range of concentrations in the antimicrobial assays among others.

Validity of the findings

Result & Discussion

Table 1 - No control description is given in “Antibacterial activity assays” and “Antibiofilm activity assays” subsections. So, the results of MIC in different extracts considered the intrinsic antimicrobial effect of the residual extract solvents? What was the range of concentrations evaluated?

Lines 286-292- “corresponding to minimal bactericidal concentration (MBC) values (Figure 3). Compound 3, salonitenolide, also was found to be effective on all the tested strains even if higher MIC100 and MBC values were detected with respect to cnicin (Figure 5). Compound 2, 11³,13-dihydrosalonitenolide, showed the lowest antibacterial activity with respect to compounds 1 and 2 with MIC100 values ranging from 0.5 to 7.5 mg/mL (Figure 4). It has also to be highlighted that, in the case of compound 2, significant differences between MIC100 and MBC values were detected on almost all the strains tested (Figure 4).” Figure 5 was cited before 4.

Also, Figures 3, 4, and 5 showed confusing data. For example:
“Figure 3. Minimal Inhibitory Concentration for cnicin (compound 1)
Minimal Inhibitory Concentration MIC100 (mg/mL) and Minimal Bactericidal Concentration MBC mg/mL) values determined for cnicin (compound 1) tested against a panel of Gram-negative and Gram-positive bacterial strains. Control samples are reported as green bars, whereas treated samples are reported as pink bars. Data represents the mean (±standard deviation, SD) of at least 3 independent experiments.”

Before the authors stated that MIC was established by “MIC100 values were determined as the lowest concentration responsible for no visible bacterial growth.” However, Figures 3, 4, and 5 showed Log (CFU/mL) and controls were added after treated samples. It should be the opposite.

Figure 4: “Figure 4. Minimal Inhibitory Concentration for 11³,13-dihydrosalonitenolide (compound 2). Minimal Inhibitory Concentration MIC100 (mg/mL) and Minimal Bactericidal Concentration MBC (mg/mL) values determined for 11³,13-dihydrosalonitenolide (compound 2) tested against a panel of Gram-negative and Gram-positive bacterial strains. Control samples are reported as green bars, whereas treated samples are reported as pink bars. Data represents the mean (±standard deviation, SD) of at least 3 independent experiments.” There are no pink bars, do you mean yellow bars?

Figure 5: “Figure 5. Minimal Inhibitory Concentration for salonitenolide (compound 3)
Minimal Inhibitory Concentration MIC100 (mg/mL) and Minimal Bactericidal Concentration MBC (mg/mL) values determined for salonitenolide (compound 3) tested against a panel of Gram-positive and Gram-negative bacterial strains. Control samples are reported as green bars, whereas treated samples are reported as pink bars. Data represents the mean (±standard deviation, SD) of at least 3 independent experiments.” There are no pink bars, do you mean red bars?

Why the authors did not compile the 3 plots in just one figure with subsections A), B), and C)?

Figure 6: “Figure 6. Antibiofilm activity of cnicin (1)
Antibiofilm activity of increasing concentrations of cnicin (1) evaluated by crystal violet assays on E. faecalis ATCC 29212, A. baumannii ATCC 17878, S. aureus ATCC 29213, S. typhimurium ATCC 14028, S. aureus MRSA WKZ-2, and E. coli ATCC 25922. Crystal violet was used to stain the formed biofilm and samples were analyzed by using a plate reader measuring the absorbance at 630 nm. Data represent the mean (±standard deviation, SD) of at least three independent experiments. Each experiment was carried out with triplicate determinations.” Please indicate that it is an Evaluation of antibiofilm activity through biomass assays explaining the conversion of the absorbance or OD measurement values in percentages.

Biomass evaluation should also be done with the extracts to compare the increment of antibiofilm activity.

Additional comments

Several shortcomings are present in this work and no acknowledgment is given by the authors.

---

## Round 0.2 · accepted · Accept

After going through the authors' responses to the reviewers' questions, I believe that the authors have well addressed these comments and the manuscript is acceptable now.

·

Basic reporting

Thank you for answering all my questions and concerns. I believe that the revised manuscript is suitable for publication endorsement.

I will just recommend you replace S. typhimurium with S. Typhimurium (in which the serovar name should not be in italics) along the text and figures/tables. It is not adequate to write it as a species name.

Also, in line 105 of the revised manuscript (pdf), the authors wrote Salmonella enterica subsp. enterica serovar Typhimurium putting all in italics. Please rectify it.

Experimental design

All questions and concerns were answered by the authors.

Validity of the findings

All questions and concerns were answered by the authors.

Additional comments

All questions and concerns were answered by the authors.